# Effect of Sodium Carboxymethyl Cellulose on Water and Salt Transport Characteristics of Saline–Alkali Soil in Xinjiang, China

**DOI:** 10.3390/polym14142884

**Published:** 2022-07-16

**Authors:** Jihong Zhang, Quanjiu Wang, Yuyang Shan, Yi Guo, Weiyi Mu, Kai Wei, Yan Sun

**Affiliations:** 1State Key Laboratory of Eco-Hydraulics in Northwest Arid Region of China, Xi’an University of Technology, Xi’an 710048, China; zhangjihong_eric@163.com (J.Z.); syy031@126.com (Y.S.); 15503635823@163.com (Y.G.); weiyimu@xaut.edu.cn (W.M.); 18291869766@163.com (K.W.); 11414014@zju.edu.cn (Y.S.); 2College of Water Resources and Architectural Engineering, Shihezi University, Shihezi 832000, China; 3Key Laboratory of Modern Water-Saving Irrigation of Xinjiang Production and Construction Corps, Shihezi University, Shihezi 832000, China; 4Key Laboratory of Northwest Oasis Water-Saving Agriculture, Ministry of Agriculture and Rural Affairs, Shihezi University, Shihezi 832000, China

**Keywords:** ionized cellulose adhesive, soil quality improvement, soil infiltration, soil water movement, salt leaching

## Abstract

The scientific use of sodium carboxymethyl cellulose (CMC) to improve the production capacity of saline–alkali soil is critical to achieve green agriculture and sustainable land use. It serves as a foundation for the scientific use of CMC to clarify the water and salt transport characteristics of CMC-treated soil. In this study, a one-dimensional soil column infiltration experiment was carried out to investigate the effects of different CMC dosages (0, 0.2, 0.4, 0.6, and 0.8 g/kg) on the infiltration characteristics, infiltration model parameters, water and salt distribution, and salt leaching of saline–alkali soil in Xinjiang, China. The results showed that the final cumulative infiltration of CMC-treated soil increased by 8.63–20.72%, and the infiltration time to reach the preset wetting front depth increased by 1.02–3.96 times. The sorptivity (*S*) in the Philip infiltration model and comprehensive shape coefficient (α) in the algebraic infiltration model showed a trend of increasing first and then decreasing with CMC dosage, revealing a quadratic polynomial relationship. The algebraic model could accurately simulate the water content profile of CMC-treated soil. CMC enhanced the soil water holding capacity and salt leaching efficiency. The average soil water content, desalination rate, and leaching efficiency were increased by 5.18–15.54%, 21.17–57.15%, and 11.61–30.18%, respectively. The effect of water retention and salt inhibition on loamy sand was the best when the CMC dosage was 0.6 g/ kg. In conclusion, the results provide a theoretical basis for the rational application of CMC to improve saline–alkali soil in arid areas.

## 1. Introduction

Human survival depends on the availability of cultivated land [1]. However, as the world’s population grows, cultivated land resources become scarce, making it difficult to meet rising food demand [2,3]. According to the statistics of the Food and Agriculture Organization of the United Nations, the world’s arable land is decreasing at the rate of 7 × 10^6^ hm^2^ per year [4]. By 2050, no new arable land resources will be added to the Earth [5]. At that time, the per capita arable land area will be only 0.15 hm^2^, posing major challenges to human society’s long-term development [6]. The saline–alkali land in Xinjiang, China, is classified as inferior soil, and it is also a valuable reserve of land resources [7]. The development and usage of Xinjiang’s saline–alkali land has become a major means of resolving the supply–demand gap in the cultivated land resources of China [8]. The sandy saline–alkali soil in Xinjiang needs to be improved in terms of soil water retention ability while inhibiting soil salt accumulation [9]. Continuously exploring innovative methods for optimal water and salt retention is critical for the development and long-term use of saline–alkali soil in Xinjiang.

Sodium carboxymethyl cellulose (CMC) is an organic compound with the chemical formula of [C_6_H_7_O_2_(OH)_2_OCH_2_COONa]_n_. It is a carboxymethylated derivative of cellulose and the most important ionic cellulose adhesive. CMC is usually an anionic polymer compound prepared by the reaction of natural cellulose with caustic soda and monochloroacetic acid, with molecular weight ranging from tens of millions to millions. Moreover, CMC is a water-soluble polymer with a high viscosity that can improve and sustain soil structure while also increasing cohesion between soil particles [10,11]. Many academics have used it to improve soil structure and water retention performance because of its wide availability, nontoxicity, and easy degradation by microorganisms in the soil [12]. Qiu et al. (2013) reported that CMC could improve soil noncapillary porosity [13]. Xi et al. (2018) found that the use of CMC increased the cohesion of aeolian sandy soil and improved the shear strength of soil, resulting in higher water retention and sand fixation, which aided in the ecological restoration of desert areas [14]. Li et al. (2017) pointed out that CMC treatment maintained a high concentration of ammonium nitrogen in the 15 cm surface layer, which improved soil erosion resistance and promoted crop growth [15]. Wu et al. (2015) demonstrated that CMC could improve the soil water holding capacity and impact soil hydraulic parameters [16]. Halidaimu et al. (2020) showed that the application of CMC reduced the soil water infiltration and restricted the evaporation process of the soil surface [17].

Despite the lack of evidence that CMC inhibits salt and improves saline–alkali soil, some researchers have begun using water-soluble polymer soil amendments such as polyacrylamide (PAM) and *γ*-polyglutamic acid to improve saline–alkali soil in various places [18,19]. PAM can combine fine soil particles into large aggregates, increasing soil porosity and creating additional pathways for soil salt leaching, which can be used for the improvement of saline–alkali soil [20,21]. Cao et al. (2017) used PAM to significantly improve the saline–alkali soil in the Jiangsu coastal reclamation area, China [22]. Alimu et al. (2019) discovered that PAM effectively improved the saline–alkali soil in Shandong coastal beaches, China [23]. Ni et al. (2020) combined the application of *γ*-polyglutamic acid and organic acid soil conditioner to significantly improve saline–alkali soil in the Yellow River Delta [24]. All of these studies demonstrated the possibility of using CMC to inhibit salt and improve saline–alkali soil.

In summary, CMC can improve the soil water holding capacity and inhibit soil evaporation by changing the soil pore structure, indicating that it has considerable potential for saline–alkali soil improvement. However, there are inadequate studies on the water and salt transport characteristics of CMC-treated soil. Therefore, we conducted an indoor one-dimensional vertical infiltration test, analyzed the effects of different CMC dosages on soil water and salt transport, established the relationship between CMC dosages and infiltration model parameters, and quantitatively analyzed its effect on the desalination of different soil layers, so as to provide a reference for the application of CMC in saline–alkali land improvement in arid areas.

## 2. Materials and Methods

### 2.1. Experimental Materials

The tested soil samples were collected from the 0–40 cm soil layer of the experimental field (86° 10′ N, 41° 35′ E) at the Bazhou Water Conservancy Administration Experimental Station in Xinjiang, China. The bulk density was determined using the ring knife method, and the recovered soil samples were air-dried before being utilized as a backup through a 2 mm sieve [25]. The mechanical composition was determined using a laser particle size analyzer (Mastersizer 2000, Marvin Instruments Co., Ltd., UK). Soil texture was classified according to international soil texture classification standards, and the basic physical properties of the soil are shown in Table 1. The water content of soil samples taken from the field after air drying for 2 days was taken as initial water content (*θ_i_*). Saturated water content (*θ_s_*) was measured by ring knife immersion method [26]. According to the method of Xu (2022) [27], the contents of major ions K^+^, Na^+^, Ca^2+^, Mg^2+^, Cl^−^, HCO_3_^−^, and SO_4_^2−^ in the soil were 0.28, 1.36, 0.32, 0.64, 2.45, 0.06, and 0.18 g/kg, respectively. Sodium chloride dominated the soil salt composition, while Cl^−^/SO_4_^2−^ levels were above 13.

### 2.2. Experimental Design

#### 2.2.1. Experimental Process

The experiment was carried out in September 2019 at the State Key Laboratory of Eco-Hydraulics in the northwest arid region of China (Xi’an University of Technology). CMC was applied at five dosages (0, 0.2, 0.4, 0.6, and 0.8 g/kg) in a mixed manner (CMC was evenly mixed with dry soil). Each treatment was repeated three times, and a total of 15 simulated infiltration tests were conducted. The experimental system mainly included a water supply device (Marriott bottle), infiltration device (organic glass column), and fixed bracket (Figure 1). The organic glass soil column was 50 cm in height and 5 cm in inner diameter; the bottom was filled with a vent with a diameter of 3 mm. The Marriott bottle mainly provided a stable infiltration head for the test, which was 50 cm in height and 5 cm in diameter. The soil sample was placed into the soil column according to a bulk density of 1.43 g/cm^3^ per 5 cm layer, with the soil layer scraped between the layers to prevent soil stratification. To avoid the occurrence of filled soil wall detachment, the area near the edge wall was repeatedly stirred and compacted with wooden rods. After filling the soil column, a layer of filter paper was placed on top of the soil to ensure uniform water infiltration and prevent splashing on the soil surface during infiltration. The water depth was kept at 2 cm, and the water level of the Marriott bottle and the wetting front depth of the soil column were recorded regularly according to a stopwatch using the principle of first dense and then sparse. The water supply was stopped when the wetting front depth reached 30 cm, absorbing the surface water immediately. Then, the soil samples were gathered at depths of 1, 5, 10, 15, 20, 25, and 30 cm.

#### 2.2.2. Analytical Methods

The drying method (105 ± 5 °C) was used to determine the soil mass water content [28], and the product of soil mass water content and bulk density was used to determine the soil volumetric water content. The dried soil samples were ground and extracted at a soil-to-water ratio of 1:5. The conductivity (EC_1:5_, mS/cm) was measured using a DDS-307 conductivity meter after standing for 8 h. Tan et al. (2017) proposed a method for calculating the soil salt content (SC, g/kg) [29].
(1)$$\mathrm{SC}=4.25\times {\mathrm{SC}}_{1:5}$$

### 2.3. Infiltration Model

In order to analyze the effect of CMC dosages on soil infiltration model parameters, we used the Philip model [30] and algebraic model [31] to investigate the infiltration characteristics of different CMC application rates. In the Philip infiltration model, cumulative infiltration is expressed as follows:(2)$$I=St^{0.5}$$
where *I* is the cumulative infiltration (cm), *S* is the sorptivity (cm/min^0.5^), and *t* is the infiltration time (min).

The algebraic model can not only adequately describe the cumulative infiltration, but also predict the water content distribution of different soil depths following accumulation infiltration [32].
(3)$$I=Z_{f}(\theta_{s}-\theta_{r})\frac{1}{1+\alpha }$$
(4)$$\theta =\left(1-\frac{Z}{Z_{f}}\right)(\theta_{s}-\theta_{r})+\theta_{r}$$
where *Z_f_* is the wetting front depth (cm), and α is the comprehensive shape coefficient of the unsaturated hydraulic conductivity and soil water characteristic curve. *θ*, *θ_s_*, and *θ_r_* are the soil water content, saturated water content, and residual water content (cm^3^/cm^3^), respectively. *Z* represents the soil depth (cm).

### 2.4. Statistical Analysis

All measured data were recorded in Excel 2019 and examined by analysis of variance (ANOVA) using SPSS 22.0 software (IBM Corp., Armonk, NY, USA). Significant differences (*p* < 0.05) between means were identified using the least significant difference (LSD) test. The determination coefficient (*R*^2^), root mean square error (RMSE), and mean absolute error (MAE) were used to evaluate the accuracy and reliability of the model simulation. A value of *R*^2^ closer to 1 indicates a higher simulation accuracy, whereas values of RMSE and MAE closer to 0 indicate a measured value closer to the simulation value. Figures were drawn using Origin 2021 software.

## 3. Results and Discussion

### 3.1. Water Infiltration: CMC Dosage Effect

The variation process of the cumulative infiltration and wetting front of the soil treated with CMC over time is shown in Figure 2. The difference in cumulative infiltration at the initial stage of infiltration was minor (Figure 2a). The impact of CMC on cumulative infiltration became apparent once the infiltration time exceeded 100 min. This was due to the fact that the soil was relatively dry at the beginning of the infiltration, and the soil matrix potential played a leading role, whereas CMC needed time to integrate with the water molecules before it could play a role [33]. The humidity of the soil surface layer increased with the infiltration period (after 100 min), and CMC fully acted on the surface soil, manifesting its cohesiveness performance and causing the cumulative infiltration of soil under different CMC dosages to gradually widen the gap. The cumulative infiltration of CMC-treated soil was lower than that of the control at the same infiltration time. When the infiltration time was 300 min, the cumulative infiltration of soil treated with 0.2, 0.4, 0.6, and 0.8 g/kg CMC decreased by 23.19%, 33.15%, 46.01%, and 41.12%, respectively, which is consistent with the experimental results of Wu et al. (2015) [16]. This was mainly because CMC molecules contain hydrophilic carboxyl and hydroxyl groups, which have a high water absorption capacity and can be fused with water molecules in the soil to form hydrogels, thereby increasing the viscosity of water and reducing the soil water infiltration rate [15]. This demonstrated that the CMC-treated soil struggled to infiltrate more water in a short time; thus, the application of CMC should be combined with a slightly greater frequency of drip irrigation. Compared with the control, the cumulative infiltration of soil treated with 0.2, 0.4, 0.6, and 0.8 g/kg CMC increased by 8.63%, 15.78%, 20.72%, and 17.75%, respectively, when the preset depth (30 cm) was attained. This was due to the fact that CMC improved the soil pore structure and increased the number of soil fine pores, thereby improving the water holding capacity of each soil layer and boosting the soil cumulative infiltration [10]. This suggested that the application of CMC could introduce more free water into the soil fine pores, which was beneficial to the full leaching of soil salinity.

The advancing speed of the wetting front gradually decreased with the increase in CMC dosage. The wetting front depth of CMC-treated soil was significantly lower than that of the control treatment at the same infiltration time. The wetting front advance distance of soil treated with 0.2, 0.4, 0.6, and 0.8 g/kg CMC decreased by 29.29%, 42.26%, 55.28%, and 50.00%, respectively, after 300 min of infiltration. In comparison to the control, the infiltration time of soil treated with 0.2, 0.4, 0.6, and 0.8 g/kg CMC increased by 1.02, 2.01, 3.96, and 3.03 times, respectively, at the preset wetting front depth (30 cm). This was mainly because the application of CMC could promote the flocculation of soil colloids, increase the number of water-stable aggregates and fine pores, prolong the soil water path, and reduce the advancing speed of the soil wetting front [24]. This is consistent with the results of Halidaimu et al. (2020), who indicated that CMC could effectively slow down the diffusion of soil water into the lower layer, keep more water in the upper soil layer, reduce the soil salt concentration in the upper layer, and effectively control the water and salt environment of the crop rhizosphere soil [17].

### 3.2. Infiltration Model Parameters: CMC Dosage Effect

The Philip and algebraic infiltration models were used to fit the measured infiltration data (Table 2). These two infiltration models had a good fitting effect, and the determination coefficients (*R*^2^) of the two models were above 0.95, indicating that the two infiltration models could successfully describe the infiltration process of CMC-treated soil.

The sorptivity (*S*) reflects the ability of the soil matrix potential to affect soil water movement [34]. The *S* increased first and then reduced as the CMC dosage was increased. The *S* was lowest when the application rate of CMC was 0.6 g/kg, which was 46.0% lower than that of the control. This is because when the CMC dosage was less than 0.6 g/kg, with the increase in CMC application amount, the flocculation of soil colloids increased, the number of soil water-stable aggregates and fine pores increased, and the water holding capacity of the soil increased. When the application rate of CMC was higher than 0.6 g/kg, excessive CMC began to plug the soil pores and reduce their connectivity. This indicated that CMC weakened the absorption capacity of soil water via the capillary force and reduced the infiltration capacity of soil water. This result is consistent with the results of Wu et al. (2015) on the water infiltration process of CMC-treated soil [16]. A quadratic polynomial was used to fit the relationship between *S* and CMC dosage (Figure 3a). The determination coefficient *R*^2^ of the fitting result was 0.984, indicating that *S* and the CMC dosage had a satisfactory quadratic polynomial relationship.

The comprehensive shape coefficient (α) for the algebraic model first decreased and then increased with the increase in CMC dosage. The α was lowest when the CMC dosage was 0.6 g/kg, which was 85.9% lower than the control. A quadratic polynomial was used to fit the relationship between α and CMC dosage (Figure 3b). The determination coefficient *R*^2^ of the fitting results was 0.993, indicating a good quadratic polynomial relationship between α and CMC dosage.

### 3.3. Accuracy Analysis of Algebraic Model to Simulate Water Content Profile of CMC-Treated Soil

The soil water content could be calculated by substituting the comprehensive shape coefficient (α) into Equation (4). The relationship between the calculated soil water content and the measured value is shown in Figure 4. The simulation results were evaluated in terms of *R*^2^, RMSE, and MAE. The calculated values of the water content profile were quite close to the measured values. The error analysis showed that the *R*^2^, RMSE, and MAE of soil water content were 0.973–0.989, 0.006–0.009, and 0.003–0.006, respectively. The main reason for the error was that the soil was isotropic and uniformly distributed in the model, which was impossible to achieve in practice [35]. Although there was some error between the simulated and measured values, according to the allowable error range given by Santhi et al. (2001), the simulation results in this study could meet the accuracy requirements and could be applied for the simulation of water infiltration of soil treated with CMC [36]. This might provide a theoretical reference for mastering the water and salt transport law of CMC-treated soil.

### 3.4. Water–Salt Distribution: CMC Dosage Effect

The distribution of water and salt in the soil profile under different CMC dosages is shown in Figure 5. The soil water content gradually decreased with the increase in soil depth (Figure 5a). Compared with the control, the average water content of soil treated with 0.2, 0.4, 0.6, and 0.8 g/kg CMC increased by 5.18%, 9.95%, 15.54%, and 12.17%, respectively. The soil water retention effect was best when the CMC dosage was 0.6 g/kg. This was mainly because CMC increased the number of water-stable aggregates and fine pores and improved the water holding capacity of the soil [33,34]. The salt content increased with the increase in soil depth (Figure 5b). The desalination effect in the 0–10 cm soil layer was obvious, while 20 cm was the boundary point between the desalination area and the salt accumulation area.

### 3.5. Salt Leaching: CMC Dosage Effect

The ratio of the difference between the initial salinity (*SC*_0_) of a certain soil layer in the desalination area and the soil salinity after infiltration (*SC*_1_) to *SC*_0_ was defined as the desalination rate (*DR*).
(5)$$DR=\frac{SC_{0}-SC_{1}}{SC_{0}}\times 100\%$$

The ratio of the difference between the desalination rate of CMC-treated soil (*DR_c_*) and the desalination rate of the control treatment (*DR_ck_*) to the *DR_ck_* was defined as the desalination intensity (*DS*).
(6)$$DS=\frac{DR_{c}-DR_{ck}}{DR_{ck}}\times 100\%$$

The ratio of the total salt leached in a certain soil layer (*S_t_*) in the desalination area to the total water flow through this soil layer (W) was defined as the leaching efficiency (*LE*, g/L).
(7)$$LE=\frac{S_{t}}{W}$$

In order to better evaluate the effect of CMC on salt leaching, the *DR*, *DS*, and *LE* of the 0–20 cm soil layer in the desalination area were calculated (Table 3). The *DR* of CMC-treated soil was significantly higher than that of the control treatment (*p* < 0.5). The *DR* first increased and then decreased with the increase in CMC dosage. The desalination intensity was the highest (57.15%) when the CMC dosage was 0.6 g/kg. In addition, the leaching efficiency reached the maximum when the CMC dosage was 0.6 g/kg, which was 30.18% higher than the control. This might be due to the existence of CMC in aqueous solution in the form of anions (i.e., significant negative charges), excluding other anions in the soil; hence, the soil salinity was lower when the upper soil profile was high [37,38].

## 4. Conclusions

The results show that although CMC decreased the wetting front advance rate of soil, it could increase the cumulative infiltration by improving the soil pore structure. The final cumulative infiltration of soil treated with 0.2, 0.4, 0.6, and 0.8 g/kg CMC increased by 8.63%, 15.78%, 20.72%, and 17.75%, respectively. With the increase in CMC dosage, the sorptivity (*S*) according to the Philip model and comprehensive shape coefficient (α) according to the algebraic model showed a trend of increasing first and then decreasing, revealing their good quadratic polynomial relationship with the CMC dosage. The algebraic model could accurately simulate the soil water content profile under CMC application. CMC increased the water holding capacity of loamy sand and improved the soil salt leaching efficiency. The desalination rate and leaching efficiency increased first and then decreased with the increase in CMC dosage. When the CMC dosage was 0.6 g/kg, the desalination rate and leaching efficiency reached the maximum, and the effect of salt leaching was best. The results fully illustrated that CMC had the ability to retain water and inhibit salt, providing a theoretical basis for the rational application of CMC to improve saline–alkali soil in arid areas.

## Figures and Tables

**Figure 1 polymers-14-02884-f001:**
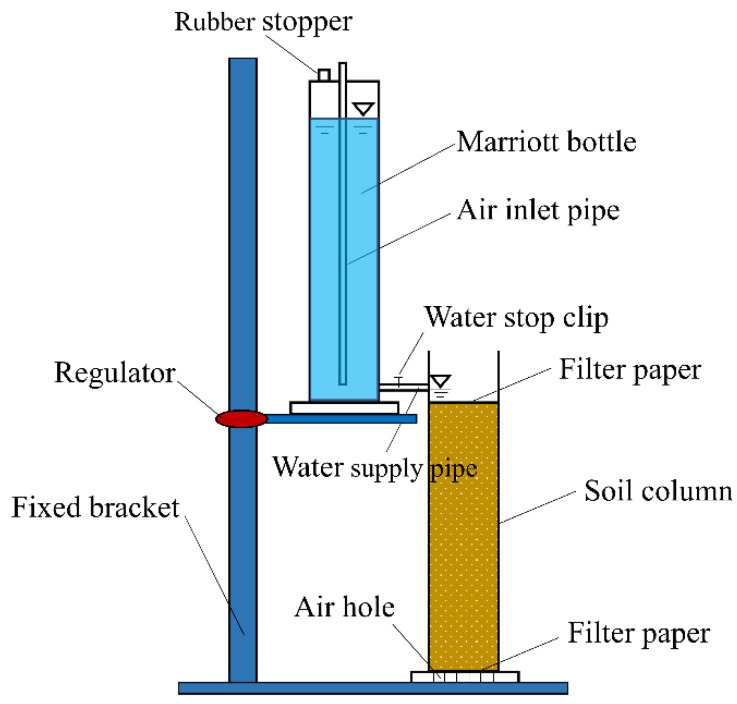
Schematic diagram of the experimental system.

**Figure 2 polymers-14-02884-f002:**
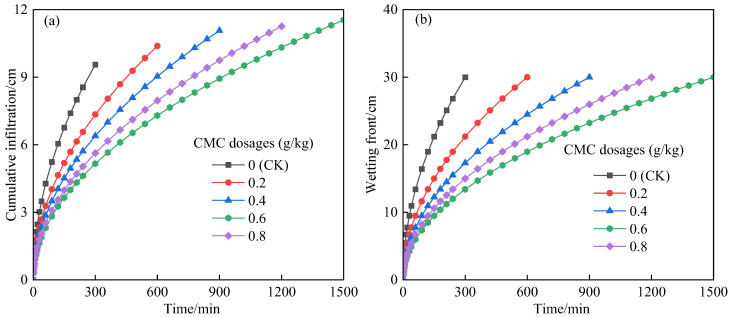
Effect of CMC dosages on soil infiltration characteristics. (**a**) Cumulative infiltration; (**b**) wetting front.

**Figure 3 polymers-14-02884-f003:**
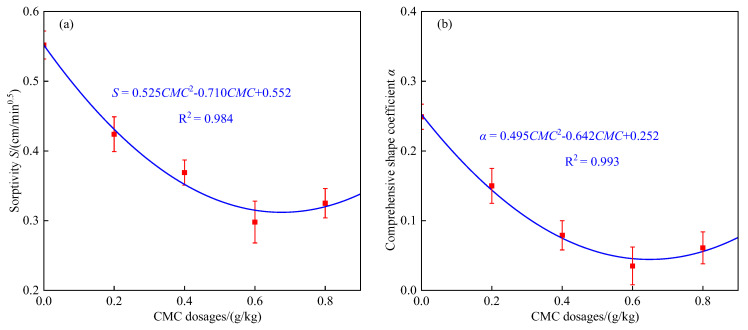
Effect of CMC dosages on infiltration model parameters. (**a**) Sorptivity *S*; (**b**) comprehensive shape coefficient (α).

**Figure 4 polymers-14-02884-f004:**
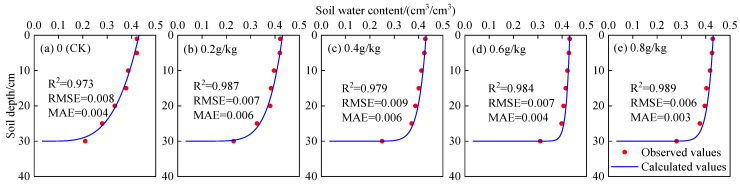
Comparison of the calculated and observed soil water content. (**a**) 0 (CK); (**b**) 0.2 g/kg; (**c**) 0.4 g/kg; (**d**) 0.6 g/kg; (**e**) 0.8 g/kg.

**Figure 5 polymers-14-02884-f005:**
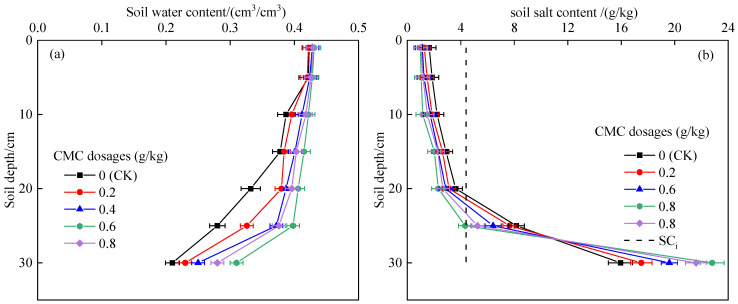
Effect of CMC dosage on soil water and salt content distribution. (**a**) Soil water content; (**b**) soil salt content.

**Table 1 polymers-14-02884-t001:** The basic physical properties of soil.

Soil Particle Composition (%)			Soil Texture	BD (g/cm^3^)	*θ_i_* (cm^3^/cm^3^)	*θ_s_* (cm^3^/cm^3^)	SC_i_ (g/kg)	pH
Sand	Silt	Clay						
83.61	15.1	1.29	Loamy sand soil	1.57	0.033	0.431	4.40	7.86

BD, *θ_i_*, *θ_s_*, and SC_i_ represent the soil bulk density, initial water content, saturated water content, and initial salt content, respectively.

**Table 2 polymers-14-02884-t002:** Infiltration model parameters under different CMC dosages.

CMC Dosages (g/kg)	Philip Model		Algebraic Model	
	Sorptivity *S* (cm/min^0.5^)	Determination Coefficient (*R*^2^)	Comprehensive Shape Coefficient (α)	Determination Coefficient (*R*^2^)
0 (CK)	0.552	0.994	0.249	0.989
0.1	0.424	0.987	0.150	0.964
0.2	0.369	0.985	0.079	0.987
0.4	0.298	0.988	0.035	0.976
0.6	0.325	0.986	0.061	0.983

**Table 3 polymers-14-02884-t003:** Effect of CMC dosage on soil salt leaching.

CMC Dosages (g/kg)	Desalination Rate (%)	Desalination Intensity (%)	Leaching Efficiency (g/L)
0 (CK)	40.07 e		5.27 e
0.1	48.59 d	21.27 d	5.89 d
0.2	53.84 c	34.37 c	6.12 c
0.4	62.97 a	57.15 a	6.87 a
0.6	56.98 b	42.20 b	6.37 b

Different letters in the same column indicate a significant difference at *p* < 0.05.

## Data Availability

The data presented in this study are available on request from the corresponding author. The data are not publicly available as the project has not been completed.

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
