# Peer review of "Effect of Sodium Carboxymethyl Cellulose on Water and Salt Transport Characteristics of Saline–Alkali Soil in Xinjiang, China"

_polymers, 2022, doi:10.3390/polym14142884_

Round 1

Reviewer 1 Report

In this work a one-dimensional soil column infiltration experiment was carried out to investigate the effects of different CMC dosages on the infiltration characteristics, infiltration model parameters, water and salt distribution, and salt leaching of saline–alkali soil in Xinjiang, China. The structure, content and the English of the manuscript are suitable and adequate. However, authors are asked to further address the following queries before acceptance:

[1] The abstract is already long, but a phrase with a methodology before presenting main results and conclusion may benefit the readering and completeness.

[2] typo: remove the paragraph point before equation (1). See also throughout the document before e.g. Eqs. (3) and (4) or Eqs (5), (6) and (7).

[3] Fig 2. The caption of the figure must present both subfigures (a) and (b) with a local description for each.

[4] Correct the latex positioning of Tab 2 on single page e.g. with the option [t] from top.

[5] Fig 3. also needs a caption for each subfigure (a) and (b), respectively.

[6] Fig 4 needs a more detailed caption for the subfigures.

[7] Idem for Fig. 5!

[8] i.e. must be italic: \italic{i.e.} in line 285 p.9

Author Response

Point 1: The abstract is already long, but a phrase with a methodology before presenting main results and conclusion may benefit the readering and completeness.

Response 1: According to the suggestion of reviewer, a phrase with a methodology before conclusion has been added.

Point 2: typo: remove the paragraph point before equation (1). See also throughout the document before e.g. Eqs. (3) and (4) or Eqs (5), (6) and (7).

Response 2: The paragraph point before equation (1), (2), (3), (4), (5), (6) and (7) has been removed.

Point 3: Fig 2. The caption of the figure must present both subfigures (a) and (b) with a local description for each.

Response 3: The local description for each subfigure has been added.

Point 4: Correct the latex positioning of Tab 2 on single page e.g. with the option [t] from top.

Response 4: The latex positioning of Table 2 has been corrected.

Point 5: Fig 3. also needs a caption for each subfigure (a) and (b), respectively.

Response 5: The caption for each subfigure (a) and (b) has been added.

Point 6: Fig 4 needs a more detailed caption for the subfigures.

Response 6: The caption for each subfigure has been added.

Point 7: Idem for Fig. 5!

Response 7: The caption for each subfigure has been added.

Point 8: i.e. must be italic: \italic{i.e.} in line 285 p.9

Response 8: It has been revised.

Reviewer 2 Report

The work "Effect of Sodium Carboxymethyl Cellulose on Water and Salt Transport Characteristics of Saline Alkali Soil in Xinjiang, China" has an interesting subject of the research.

The presented results ca be useful!

However, the presented data must be support to adequate materials characteristics.

For this reason authors must the authors must present the most accurate (median) chemical composition of the soil samples.

Also, more details about the characteristics of the polymer samples are needed.

In the other hand, the correlations between polymer ans soils samples must presented.

Author Response

Point 1: Authors must the authors must present the most accurate (median) chemical composition of the soil samples.

Response 1: According to the method of Xu (2022) [27], the contents of major ions K+, Na+, Ca2+, Mg2+, Cl, HCO3, and SO42− in the soil were 0.28, 1.36, 0.32, 0.64, 2.45, 0.06, and 0.18 g/kg, respectively. Sodium chloride dominated the soil salt composition, while Cl/SO42− levels were above 13.

Point 2: Also, more details about the characteristics of the polymer samples are needed.

Response 2: The details about the characteristics of the polymer samples have been added. “Sodium carboxymethyl cellulose (CMC) is an organic compound with the chemical formula of [C6H7O2(OH)2OCH2COONa]n. It is a carboxymethylated derivative of cellulose and the most important ionic cellulose adhesive. CMC is usually an anionic polymer compound prepared by the reaction of natural cellulose with caustic soda and monochloroacetic acid, with molecular weight ranging from tens of millions to millions. Moreover, CMC is a water-soluble polymer with a high viscosity that can improve and sustain soil structure while also increasing cohesion between soil particles [10,11].”

Round 2

Reviewer 2 Report

The revised manuscript "Effect of Sodium Carboxymethyl Cellulose on Water and Salt Transport Characteristics of Saline Alkali Soil in Xinjiang, China" respond to all my corrections request.

The work can be considered for the publish.